# Speaking to a common tune: Between-speaker convergence in voice fundamental frequency in a joint speech production task

**Vincent Aubanel**[1]*, **Noël Nguyen**[2,3]

**1** University of Grenoble Alpes, CNRS, GIPSA-lab, Grenoble, France, **2** Aix Marseille Univ, CNRS, LPL, Aix-en-Provence, France, **3** Aix Marseille Univ, Institute for Language, Communication and the Brain, Marseille, France

* vincent.aubanel@gipsa-lab.fr

**Data Availability Statement:** The data are available at https://doi.org/10.5281/zenodo.3630439.

**Funding:** This work has been conducted with the financial support of the French National Research Agency (anr.fr) and the Excellence Initiative of Aix-

## Abstract

Recent research on speech communication has revealed a tendency for speakers to imitate at least some of the characteristics of their interlocutor's speech sound shape. This phenomenon, referred to as phonetic convergence, entails a moment-to-moment adaptation of the speaker's speech targets to the perceived interlocutor's speech. It is thought to contribute to setting up a conversational common ground between speakers and to facilitate mutual understanding. However, it remains uncertain to what extent phonetic convergence occurs in voice fundamental frequency ($F_0$), in spite of the major role played by pitch, $F_0$'s perceptual correlate, as a conveyor of both linguistic information and communicative cues associated with the speaker's social/individual identity and emotional state. In the present work, we investigated to what extent two speakers converge towards each other with respect to variations in $F_0$ in a scripted dialogue. Pairs of speakers jointly performed a speech production task, in which they were asked to alternately read aloud a written story divided into a sequence of short reading turns. We devised an experimental set-up that allowed us to manipulate the speakers' $F_0$ in real time across turns. We found that speakers tended to imitate each other's changes in $F_0$ across turns that were both limited in amplitude and spread over large temporal intervals. This shows that, at the perceptual level, speakers monitor slow-varying movements in their partner's $F_0$ with high accuracy and, at the production level, that speakers exert a very fine-tuned control on their laryngeal vibrator in order to imitate these $F_0$ variations. Remarkably, $F_0$ convergence across turns was found to occur in spite of the large melodic variations typically associated with reading turns. Our study sheds new light on speakers' perceptual tracking of $F_0$ in speech processing, and the impact of this perceptual tracking on speech production.

## Introduction

In spoken-language interactions, recent work has revealed that speakers tend to imitate their interlocutor's own way of speaking (see [1] for a recent review). This phenomenon, referred to

Marseille University (A*MIDEX, amidex.univ-amu. fr) (Grant Agreements no. ANR-08-BLAN-0276-01, ANR-16-CONV-0002 (ILCB) and ANR-11-LABX-0036 (BLRI)), and of the European Research Council (erc.europa.eu) under the European Community's Seventh Framework Program (FP7/2007-2013 Grant Agreement no. 339152, "Speech Unit(e)s", J.-L. Schwartz PI). The funders had no role in study design, data collection and analysis, decision to publish, or preparation of the manuscript.

**Competing interests:** The authors have declared that no competing interests exist.

as phonetic convergence, entails a moment-to-moment adaptation of the speaker's speech targets to the perceived interlocutor's speech patterns. It is thought to contribute to setting up a conversational common ground between speakers, to facilitate mutual understanding, and to strengthen social relationships [2]. In addition, phonetic convergence has been found to persist after the interaction has ended [3], and this provides evidence for the emerging view that words' spoken forms in the mental lexicon continuously evolve throughout the speaker's lifespan under exposure to speech produced by other speakers.

In the present work, we focused on voice fundamental frequency ($F_0$), a central dimension of speech, as a conveyor of both linguistic information (through intonation patterns, as well as lexical tones in tone languages, in particular) and communicative cues associated with the speaker's age, gender, and/or emotional state. Our main objective was to contribute to better characterizing the size of convergence effects in $F_0$ in both the temporal and frequency domains. More specifically, we aimed to experimentally determine whether, and if so to what extent, convergence in $F_0$ between human speakers extends across speakers' turns. We also sought to establish how accurately speakers may imitate changes in their partner's $F_0$ that are both limited in magnitude and spread over large intervals.

Previous studies have examined potential between-speaker convergence effects in $F_0$ by means of direct, acoustic measures [4–15], indirect, perceptual evaluations performed by listeners [16], or both [1, 17–19]. The results, however, have shown important discrepancies both across and within studies, as to whether convergence occurs or not, and if so to what extent. Many of these studies have employed a repetition task, which entails participants repeating a series of isolated vowels [4, 9], nonwords [7], words [1, 6, 17, 19], or sentences [8, 20, 21] previously recorded by one or several model speaker(s) and played out to the participants. Using this approach, [4] and [9] have provided acoustic evidence for $F_0$ convergence in the repetition of vowels, with a larger effect in [4] than [9]. [17] have reported a small but significant acoustic convergence effect in $F_0$ in single-word shadowing, a finding consistent with Goldinger's often cited, albeit unpublished early study referred to in [6]. In a VCV (/aba/) repetition task, however, [5] found that convergence in $F_0$ occurred to a small degree when participants were presented with both the audio and video recordings of the model speaker, but not in the audio-only condition. [7] had German-speaking healthy participants repeat nonwords in different tasks that included delayed repetition and shadowing. Participants showed an $F_0$ convergence effect towards the model speaker in the delayed repetition task but not in the shadowing task. Single-subject analyses revealed that in delayed repetition, the effect was significant for 3 participants only out of 10. The absence of $F_0$ convergence in the shadowing task was attributed by the authors to an overall increase in $F_0$ resulting from an enhanced speaking effort in the shadowing compared with the delayed repetition task. In a recent, single-word shadowing study [18], participants did not display a consistent trend toward acoustic convergence in $F_0$. Likewise, Pardo and colleagues [1, 19] did not find acoustic evidence for convergence in $F_0$ in their large-scale studies using single-word shadowing.

It is difficult to pinpoint what may be the origin of the disparities in the occurrence and extent of convergence effects in $F_0$ in the abovementioned studies, given the vast array of differences that these studies show at the methodological level. These differences include the number of model speakers (from one speaker, e.g. [7, 17], to 20 speakers in [19]), the number, phonological make-up and lexical status of the items used as stimuli, and the index employed to characterize convergence from the $F_0$ measures, among other features. One may note, however, that three ([4, 7, 9]) of the studies in which convergence in $F_0$ was observed appear to share one characteristic that we do not find in other studies. In [4], [7], and [9], the experimenters made $F_0$ in the stimuli vary in a systematic way, either through resynthesis ([4, 7]) or by selection of a set of $F_0$ values ([9]) in the material recorded by the model speaker(s).

Systematic variations in $F_0$ in the stimuli may have facilitated the emergence of convergence effects, compared with stimuli in which the range of $F_0$ variations was not controlled.

Convergence effects in repetition tasks have also been subject to perceptual evaluations, in conjunction with acoustic analyses [1, 17–19] or in an independent way [16]. When the participants' task is to repeat (non-)words or shorter linguistic units, perceptual evaluations are most frequently carried out by means of an AXB classification test, in which listeners are asked to determine whether the participant's shadowed version of a word (stimulus A) sounds more similar to the model speaker's version (stimulus X) compared with the participant's baseline version of that word (stimulus B, with stimuli A and B counterbalanced across trials). Because the listeners' perceptual judgments are necessarily holistic, the potential influence of $F_0$ in these judgments is difficult to disentangle from that of other acoustic parameters. In [16], however, $F_0$ was artificially manipulated independently of other parameters, by being equated across the A, X and B stimuli in the AXB test. The results showed that shadowed words were more often correctly perceived as better imitations of the model speaker's words for the original than for the equated-$F_0$ stimuli, and were therefore indicative of $F_0$ being a salient cue to imitation in single-word shadowing (see [16], footnote 4).

Work has also been carried out on potential convergence effects in tasks that involve pairs of participants speaking in a turn-taking fashion, and performed by one speaker in conjunction with another human speaker or an artificial agent. This includes conversational interactions, but also interactive verbal games (e.g., [22]), or joint reading tasks as in the present piece of work, among other examples. Gregory, Webster and colleagues conducted a series of acoustic studies [12–15, 23, 24] on dyadic interviews and dyadic conversations, which were all carried out according to the same general design. Recordings for each participant were divided into a number of excerpts equally spaced over the duration of the interaction, and a long-term average spectrum (LTAS) was computed across the low-frequency range for each excerpt. At each temporal division, the LTAS for each speaker was then compared to that of her/his interlocutor and that of the other speakers. Gregory and colleagues recurrently showed that correlations were higher for actual pairs of speakers (that had actually interacted with each other) than for virtual pairs. These findings have been taken as providing strong support for convergence in $F_0$ in conversational interactions. However, caution may be required in the interpretation of these results, due to the lack of information on different methodological and technical aspects that are central to accurately analyzing $F_0$. In particular, both sampling frequency and duration of excerpts were unspecified, as was the participants' gender in [12] (for the Arabic speakers) and [15]. In addition, whereas the LTAS was focused on a narrow low-frequency band (62-192 Hz) in [12], it was extended up to 500 Hz in [13–15], and this may have resulted in the LTAS incorporating spectral components above $F_0$, such as $F_1$ in non-low vowels in male speakers. It is also difficult to ascertain that LTAS correlations have not been affected by variations in the recording conditions across interviews (which would tend to mechanically make the correlations higher for actual compared with virtual pairs), in [13] for example. In a more recent work, [11] directly measured mean $F_0$ values associated with their participants' speaking turns in conversational exchanges with a virtual agent, and found a convergence effect in their participants towards the virtual agent's pre-recorded voice. [11] also explored their participants' potential tendency to converge towards the virtual agent in $F_0$ changes across turns. The participants' mean $F_0$ appeared to vary across turns in a periodic fashion that mirrored the periodic pattern contained in the model speakers' recorded voices, as implemented in the virtual agents.

In the present work, we asked whether, and if so to what extent, two speakers converge towards each other with respect to variations in $F_0$ in a scripted dialogue. Pairs of speakers jointly performed a speech production task, in which they were asked to alternately read aloud

a written story divided into a sequence of short fragments, or reading turns. The task was conceived with a view to studying convergence in $F_0$ in the framework of the novel experimental approach to sensori-motor integration and cognition known as joint action [25, 26]. We devised an experimental set-up that allowed us to manipulate the speakers' $F_0$ in real time and with high accuracy across turns, in a way which bears similarities with the paradigm employed by Natale in his early study [27] on convergence in vocal intensity in dyadic communication. Both our experimental design and convergence measures were specifically conceived with a view to disentangling genuine convergence effects in $F_0$ changes across turns, from similarities between speakers in $F_0$ patterns as a by-product of the fact that speakers employ shared sentence or discourse structures.

Studies on modified auditory feedback where the formants [28] or $F_0$ [29, 30] is altered in real time have found an automatic response in a direction opposite to that of the perturbation (although same-direction responses can also happen, see [31]). This behavior is accounted for by theoretical models which assume that an internal simulation of the speakers' auditory target is compared with the actual auditory feedback and issue correcting commands to the motor system when a mismatch is detected [32, 33]. This framework could explain why speakers perform with outstanding accuracy in speech tasks such as shadowing [16] or synchronous speech [34]. We hypothesised that, because $F_0$ is a core dimension of communicative behavior, speakers respond to $F_0$ modifications in their interlocutor's speech by tending to imitate these modifications.

## Materials and methods

This study was approved by the Ethics Committee of Aix-Marseille University.

### Participants

Sixty-two female native speakers of French, all undergraduate students at Aix-Marseille University and from 18 to 47 years old, took part in the experiment. We chose our sample size following Sato et al.'s [9] study, in which $F_0$ convergence effects were observed with cohorts of 24 participants exposed to $F_0$ values varying from 196 to 296 Hz (for female voices), that is, a variation of 714 cents. Given our planned variation of 400 cents, that is, about half of that in [9], we estimated that we would need at least twice as many participants to observe an $F_0$ convergence effect, but set the number of participants to 62 to be on the safe side. One pair of participants had to be discarded from the analyses owing to faulty recordings of the audio signals, leaving 60 participants in the test sample. Post-hoc analyses confirmed that this sample size was adequate to observe the expected convergence effect with our planned design (see Results).

Whether there are variations in the amount of between-speaker phonetic convergence as a function of speaker gender has been a matter of debate. In an often-cited work, [35] found that female shadowers converged towards the model speaker to a greater extent than male shadowers. However, more recent studies with a focus on gender-related differences in phonetic convergence (e.g., [1, 3, 17, 36–44]) have provided results that were inconsistent in that respect. In Pardo's seminal study [44] on phonetic convergence in conversational interactions, for example, convergence was found to be greater for male than for female speakers. In another, large-scale study, Pardo et al. [44] found no difference in the amount of phonetic convergence depending on speaker gender, whether in their conversational interaction task or in their speech shadowing task. Because gender was not a focus of interest in our study, we chose to only have female participants, as in [5, 45–49], for technical reasons explained below.

Recruitment and testing were made in accordance with the standard procedures of Aix-Marseille University at the time of recruitment. All participants provided written informed consent. They were recruited in pairs, with the requisite that a) they had no auditory, speech production or reading disorder known to them, and b) that pair members already knew each other and had an age difference of no more than ten years. Fulfillment of these criteria was established by means of a questionnaire filled in by the participants prior to the experiment.

Both the familiarity and age difference criteria were expected to facilitate coordination between participants in the reading task. Duration of acquaintance ranged from a number of weeks (6 pairs) to several months (14 pairs) or years (9 pairs). Age difference was lower than 3 years for most (26) pairs and did not exceed 8 years.

## Procedure

We asked pairs of participants to perform an alternate reading task. This entailed participants alternately reading aloud a written short story divided into a fixed number of reading turns ($N$ = 74, see below).

Each pair of participants was given a general introduction to the study by the experimenter in a control room, as well as written instructions. The two participants were then randomly assigned and dispatched to separate sound-isolated booths (A and B), where each of them was equipped with a C520 Sennheiser headset microphone and a pair of HD202 Sennheiser closed headphones. The participants' positioning in different booths ensured that each participant's voice was conveyed to the other participant through this electronic communication channel only, and that aerial sound transmission between participants was blocked. Further instructions by the experimenter were also transmitted through the communication equipment, from the control room. Participants first had to read silently the text they were to use in the alternate reading task, to familiarize themselves with it. They then did a practice session together, with a different, short text. Following this, they jointly performed two repetitions of the alternate reading task. The average duration of each repetition was 4 min and 33 s across the 32 pairs of participants. In all, the experiment lasted around 30 minutes. The experimental set-up is shown in Fig 1.

## Materials

The text used in the experiment was a simplified version of a technical notice for installing a wooden floor, chosen for its neutral style. It contained 804 words and was split into 74 turns, each from 6 to 13 words long, with turn boundaries placed within but not across sentences. This was done to avoid participants making long pauses between turns, and to favor prosodic

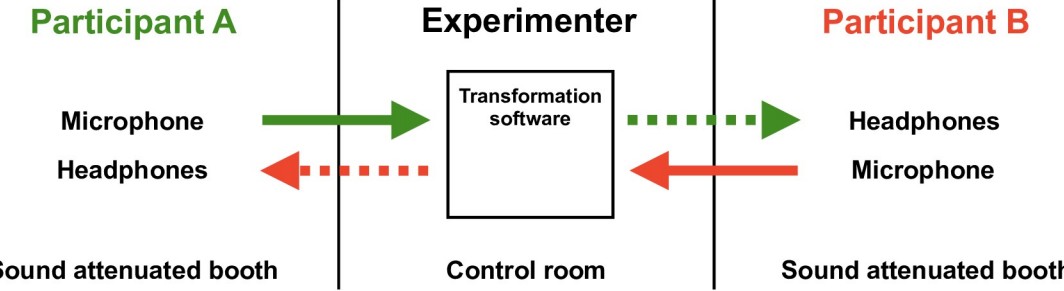

**Fig 1. Experimental set-up.** Participants are seated in individual booths and communicate with each other through microphones and headphones, while an experimenter operates the $F_0$ transformation software in the control room. Dashed lines indicate $F_0$-transformed voice.

continuity from one participant's turn to the other participant's one. The text was printed in two versions, one for participant A with odd-number turns in bold face and even-number turns in gray color, and the opposite pattern for speaker B (see S1 Text for the full text including turn segmentation).

## Experimental design

Unbeknownst to both of the participants, and using an experimental device that was placed in the control room along the communication channel between them, we artificially shifted the participants' F$_0$ from one turn to the next, by a value determined at each turn according to the following sinusoidal function:

$$\tau(t) = A \sin(\omega t + \phi), \quad t = 0, \ldots, 73 \tag{1}$$

where $A$ is the amplitude of the transformation and was set to 200 cents (2 semitones), $t$ is an index associated with the reading turns 1 to 74, $\omega$ is the angular frequency and was set to $2\pi/74$ for $\tau$ to achieve one complete cycle over the sequence of reading turns, and $\phi$ is the phase angle, set to either 0 or $\pi$, as detailed below. The F$_0$ transformation value $\tau$ was set before the beginning of each reading turn by the experimenter. The long period and limited amplitude of the transformation were both chosen so that the participants did not notice that their partner's voice had been artificially manipulated. The maximal value of $\tau$ between two consecutive turns in a given speaker, was about 34 cents, i.e., 1/6 tone, and this made it unlikely for the other speaker to detect that change, all the more so since that speaker had to produce a turn herself in between.

To assess the extent to which participants reproduced each other's shifts in F$_0$ across turns, we asked participants to perform the task twice. In one reading, the phase angle of the transformation function was 0 (hereafter, 0-phase condition). In the other reading, the phase angle was $\pi$ ($\pi$-phase condition). The order of the 0-phase and $\pi$-phase readings was counterbalanced across pairs of participants. We then computed, for each participant and each turn, the difference $\delta$ in the median of the untransformed F$_0$ values between the 0-phaseand $\pi$-phaseconditions, as follows:

$$\delta(t) = \tilde{F}^0_{0 \text{ untransf}}(t) - \tilde{F}^\pi_{0 \text{ untransf}}(t) \tag{2}$$

where $\tilde{F}^0_{0 \text{ untransf}}(t)$ and $\tilde{F}^\pi_{0 \text{ untransf}}(t)$ are the median of the untransformed F$_0$ values for turn $t$ in the 0-phaseand $\pi$-phaseconditions respectively. Because our goal here was to characterize F$_0$ patterns in the speech waveform as produced by the participants, both median values related to the participants' untransformed speech.

If we assume that each participant tends to reproduce the shifts in F$_0$ to which they are exposed in their partner's speech, as heard through the voice-transformation system, $\delta$ should mirror the variations of $\tau$ in the $\pi$-phase condition as subtracted from $\tau$ in the 0-phase condition. That is, $\delta$ should display a sinusoidal shape with a period of 74 turns and a phase angle of 0. Note that $\delta$ is computed as a difference in F$_0$ between two readings of the same text by the same two participants. As a result, $\delta$ is expected to mostly reflect the participants' degree of convergence towards the F$_0$ movements related to $\tau$ in their partner's voice, and to be little sensitive to the prosodic variations associated with specific portions of the text, specific reading style of participants, or both. These variations should tend to be abstracted away in the calculation of the F$_0$ difference between the two readings of the text by the same participant. The values of the transformation function in the two reading conditions is shown in Fig 2.

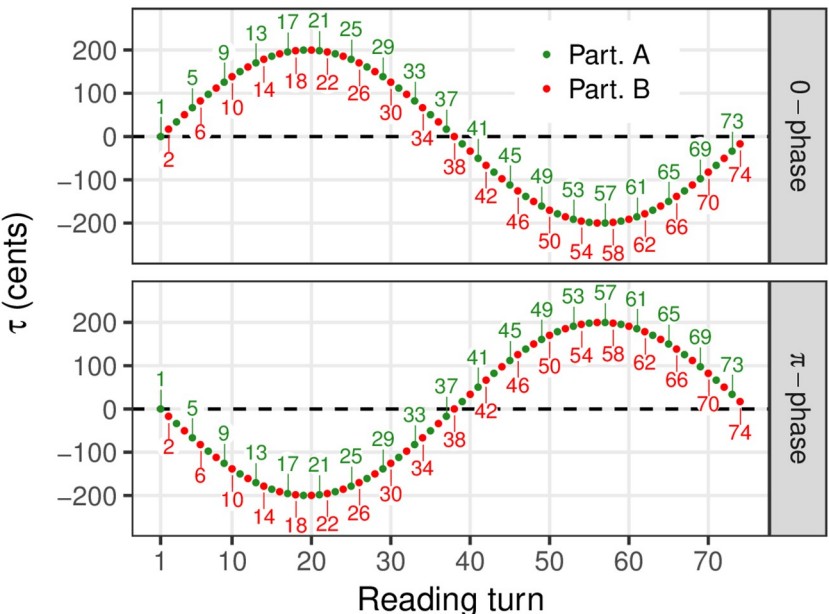

**Fig 2. $F_0$ transformation values for the two repetitions of the task.** Top: 0-phase condition. Bottom: $\pi$-phase condition.

## Voice fundamental frequency transformation

The real-time voice transformation system was implemented using the Max5 software (Cycling74). It consisted of a graphical interface that allowed us to interactively apply the $F_0$ transformation to either of the two participants' channels (see S1 Fig), and to more generally control the experimental procedure, including playing pre-recorded instructions. The voice transformation module used the phase vocoder `supervp.trans~` [50] provided within the real-time sound and music processing library IMTR-trans. Preliminary tests showed that the quality of the voice transformation was higher for female than for male speakers, owing to the fact that the higher mean $F_0$ in female voices makes it possible to use shorter analysis/resynthesis time windows. This is the reason why we recruited female participants only.

## Data analysis

For each participant, $F_0$ values were extracted every 10 ms in both the untransformed and the transformed speech recordings, as produced by the participant and heard by the participant's partner respectively. We employed a two-pass procedure (see [51]) using the Praat software [52] to minimize octave jumps and other detection errors: first, an automatic detection was performed using maximal $F_0$ register limits (75 – 750 Hz). These limits were then manually adjusted on the basis of a visual inspection of the detection results for each participant and the new, speaker-specific, limit values were used for the second and final automatic detection pass.

   The temporal location of the boundaries between consecutive reading turns was established by a silence-detection semi-automatic procedure, followed by a visual check and adjustments when necessary using a signal editor. For each turn, we then took the median $F_0$ value in the channel of the participant that had spoken during that turn, in both the untransformed and transformed speech recordings.

   To estimate to what extent the sinusoidal pattern introduced in $\tau$ (Eq 1) can be found in $\delta$ (Eq 2), we fitted a sinusoidal function to the $\delta$ data series and sought to estimate the target

parameters $A$, $\omega$ and $\phi$ from Eq 1 using nonlinear least-squares regression (function `nls()` from the R package `stats` [53]).

## Results

Pairs of participants accomplished the joint reading task in a smooth and fluent way, as indicated by the short lag (mean duration: 219 ms, SD: 253 ms) between each reading turn and the following one.

As verified during a debriefing with the experimenter that followed the experiment, none of the participants noticed that the voice of their partner had been artificially modified. As the transformations made to each participant's voice could be heard by the participant's partner but not by the participant herself, none of the participants reported that their own voice had been artificially modified either.

The accuracy of the voice transformation software was evaluated by calculating the difference between the measured transformed $F_0$ values ($F_{0\ transf}$) and their expected values, estimated by the measured untransformed $F_0$ values shifted by $\tau$ ($F_{0\ untransf} + \tau$). The distribution of the difference was highly leptokurtic (kurtosis value of 414.0), with more than 92% of the measured points lying within ±20 cents of the expected values, indicating that the voice transformation system was highly accurate.

### Between-speaker convergence in $F_0$ shifts across turns

Between-participant imitation in turn-wise $F_0$ transformation should cause $\delta$ (see Eq 2) to follow a sinusoidal pattern across turns, with the same period (74 turns) as that of the applied transformation $\tau$, and a zero phase angle.

A sinusoidal function was fitted to $\delta$ to determine the period, amplitude and phase which allowed that function to best account for the variations shown by $\delta$ across turns. We used nonlinear least-squares regression with initial conditions set to $C = 0$ cents, $A = 40$ cents, $T = 74$ turns, and $\phi = 0$ to estimate the coefficients of the model. Coefficients were estimated to $C = 7.06$ cents, $A = 24.11$ cents, $T = 75.88$ turns and $\phi = -0.68$, i.e., $-8.11$ turns (all $p < 0.001$). The resulting fit is shown in Fig 3. This indicates that, in both the 0- and $\pi$-phase conditions, participants converged towards each other by exhibiting $F_0$ variations across turns that followed a single-cycle sinusoid, with a delay of 8.11 turns with respect to the transformation applied, that is, 4.06 turns heard by each participant, and an amplitude of 12.06 cents on average over the two repetitions of the task.

A post-hoc evaluation of the replicability of the model coefficients' significativity was conducted by generating new data using the estimated model coefficients and a random error term with mean and standard deviation equal to that of the estimated residual standard error. Out of 1000 simulated datasets, the $C$, $A$, $\omega$ and $\phi$ coefficients were significant 97.2, 100, 100 and 98.1% of the time respectively. We take this high degree of replicability as a confirmation of the adequacy of our participant sample size (see Methods).

### Between-speaker convergence in mean $F_0$

We examined to what extent participants converged towards each other in mean $F_0$, by calculating the correlation in mean $F_0$ across pairs of participants. Over the entire duration of the task, this correlation was found to be significantly positive ($r = 0.45$, $p < 0.02$, see Fig 4a). When computed for each successive pair of turns (associated with participant A and B respectively), the correlation reached its greatest positive value at the beginning of the task, decreased over the first reading (slope of linear regression = $-0.003$, $p < 0.001$), and remained stable for

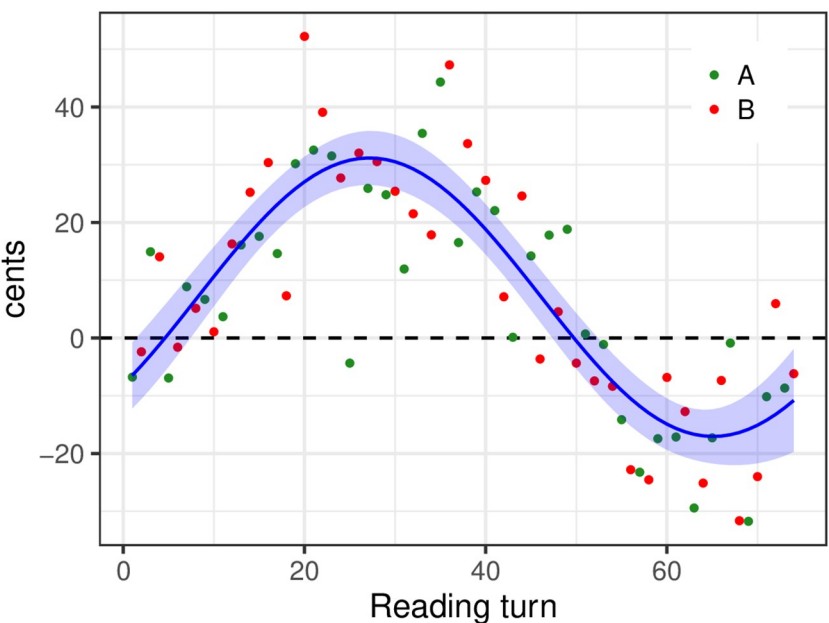

**Fig 3. Between-speaker convergence in $F_0$ shifts across turns.** $\delta$ measure: Difference between 0-phase and $\pi$-phase condition in median $F_0$ for each participant and each turn. Sinusoidal fit is shown in blue, with 95% confidence interval in light blue.

the second reading ($p = 0.43$). This was true regardless of whether participants started with the 0- or $\pi$-phase condition (see Fig 4b).

In addition, we asked to what extent the participants' tendency to imitate each other's perceived shifts in $F_0$ across turns, as measured by $\delta$, was related to how close participants were to each other in mean $F_0$ value. To answer this question, we focused on turns 19 to 22, a selection determined as the longest turn sequence where $\delta$ was found to significantly differ from 0, as evaluated by uncorrected independent t-tests on each turn. Fig 4c shows the average $\delta$ in that interval as a function of the absolute difference in grand average $F_0$ between participants. We found a significantly negative correlation between these two dimensions ($r = -0.30$, $p < 0.02$), showing that co-participants who were closer to each other in mean $F_0$ tended to more closely imitate each other's perceived shifts in $F_0$ from one turn to the next.

## Predictability of $F_0$ across turns

To further characterize the turn-by-turn dynamics of convergence in $F_0$, we evaluated to what extent the participants' median $F_0$ at each turn could be predicted from $F_0$ values in preceding turns. We performed three linear mixed-effect analyses, each predicting the participants' median untransformed $F_0$ value at turn $t$. For model $m1$, the predictor was the median transformed $F_0$ value at turn $t - 1$, i.e., the transformed $F_0$ value of the participant's partner as heard by the participant. For model $m2$, the predictor was the median untransformed $F_0$ value at turn $t - 2$, i.e., the untransformed $F_0$ of the participant's own preceding turn as heard by the participant through auditory feedback. The predictors for $m3$ were a combination of the two predictors of $m1$ and $m2$. We included for all three models the same random effect structure, obtained by increasing the complexity of the structure until adding a term did not significantly increase the explained variance. The random effect structure consisted of an intercept and a random slope by turn for the first predictor, and an intercept and a random slope by participant for both predictors. Data were transformed to $z$-scores by participant prior to modeling

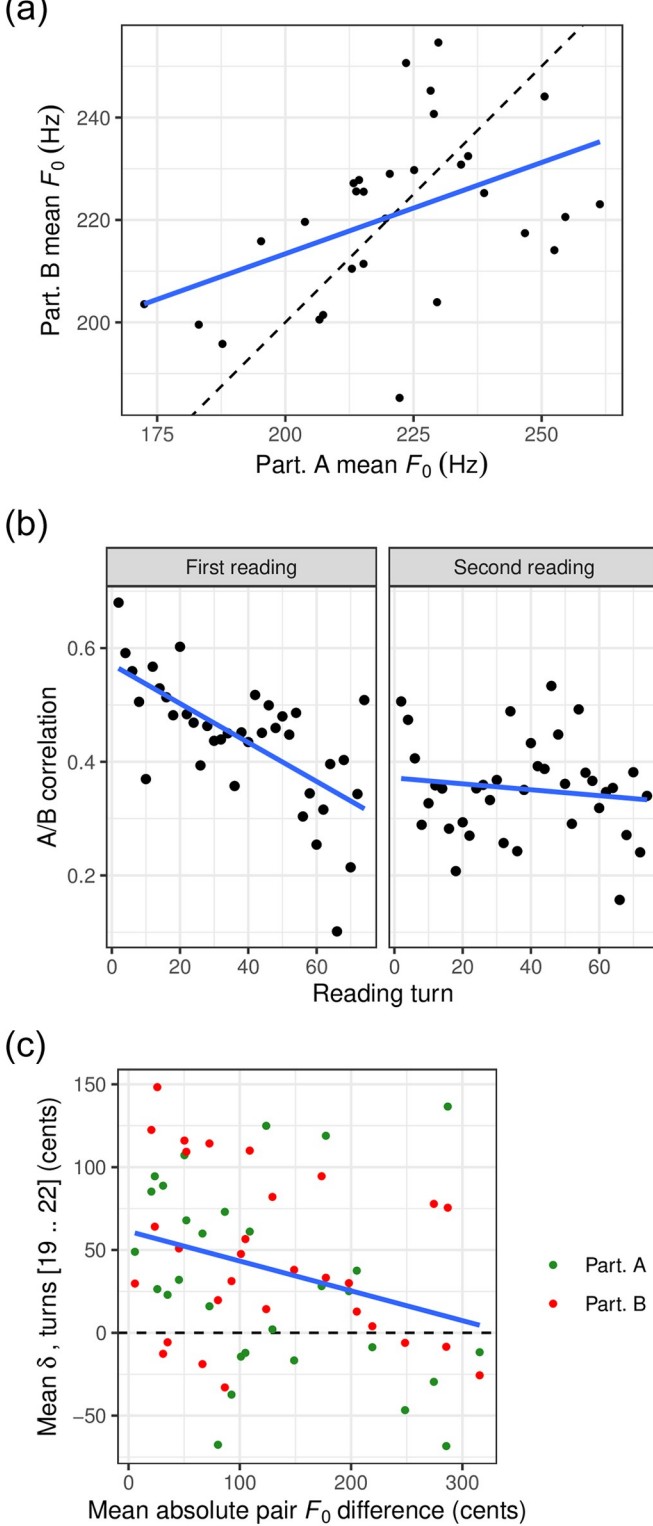

**Fig 4. Between-speaker convergence in mean $F_0$.** In each panel, the regression line is shown in blue. (a) Global A/B $F_0$ correlation: mean $F_0$ of participant A as a function of mean $F_0$ of participant B over the total duration of the task. The dashed line represents a hypothetical correlation of 1. (b) A/B correlation in $F_0$ (as in (a)) for successive pairs of turns. (c) Mean $\delta$ in turns 19 to 22 as a function of the overall $F_0$ difference of the pair members.

**Table 1. Output of linear mixed-effect modeling of the turn median $F_0$.** $\tilde{F}_0(t)$ refers to the median of the $F_0$ value in turn $t$. Columns show, from left to right: the model ID, the predictors, random effects standard deviations for terms: (a) and (b): intercept and $\tilde{F}_{0\,\text{transf}}(t-1)$ resp. by turn, (c), (d), (e): intercept, $\tilde{F}_{0\,\text{transf}}(t-1)$ and $\tilde{F}_{0\,\text{untransf}}(t-2)$ resp. by participant, (f): residuals, and Akaike's Information Criterion. To the right of the vertical separator are the results of an ANOVA between the first two models and $m3$.

| Model | Predictors | Random effects Standard Deviation | | | | | | AIC | ANOVA with $m3$ | |
|---|---|---|---|---|---|---|---|---|---|---|
| | | **(a)** | **(b)** | **(c)** | **(d)** | **(e)** | **(f)** | | $\chi^2$ | $p$ |
| $m1$ | $\tilde{F}_{0\,\text{transf}}(t-1)$ | 0.37 | 0.01 | 0.55 | 0.30 | 0.09 | 0.63 | 8891 | 87.52 | $< 2.2e\text{-}16$ |
| $m2$ | $\tilde{F}_{0\,\text{untransf}}(t-2)$ | 0.38 | 0.01 | 0.46 | 0.12 | 0.12 | 0.63 | 8827 | 23.71 | 1.122e-06 |
| $m3$ | $\tilde{F}_{0\,\text{transf}}(t-1) + \tilde{F}_{0\,\text{untransf}}(t-2)$ | 0.38 | 0.01 | 0.47 | 0.12 | 0.08 | 0.63 | 8805 | – | |

to avoid numerical convergence issues due to difference in intra- vs. inter-pair variation. Table 1 summarizes the three models' fits to the data, as well as the results of an ANOVA between the first two models and the third one in which they are both nested.

We found that when tested separately, the factors associated with $m1$ and $m2$ were significant predictors of the median $F_0$ at turn $t$, and that the combination of the two factors provided a significantly better fit than either of the factors taken separately. This suggests that in joint reading, the median $F_0$ produced by participants during a turn depends on both their own median $F_0$ in their preceding turn and their interlocutor's average median $F_0$ as just heard in the immediately preceding turn.

## Discussion

This study first demonstrates that, in a joint reading task, the two speakers tend to imitate each other's changes in $F_0$ across turns that are both limited in amplitude and spread over large temporal intervals. In our experimental set-up, the shift we introduced in each speaker's $F_0$ between two of their consecutive reading turns was always smaller than one-sixth of a tone. The observed between-speaker convergence in $F_0$ shifts across turns shows that, at the perceptual level, speakers monitor slow-varying movements in their partner's $F_0$ with high accuracy and, at the production level, that speakers exert a very fine-tuned control on their laryngeal vibrator in order to imitate these $F_0$ variations. Remarkably, $F_0$ convergence across turns was found to occur in spite of the large melodic variations typically associated with reading. Indeed, we found that the average $F_0$ range of a turn, measured as the mean difference between the turn maximum and minimum values, was 10.94 semitones ($SD = 3.00$) across participants, close to one octave, and was therefore much larger than the one-sixth of a tone shift between reading turns in each speaker.

Our results also indicate that speakers tended to converge towards each other in mean $F_0$, a tendency that was found to establish itself from the beginning of the sequence of reading turns. It is important to note that convergence in mean $F_0$, on the one hand, and convergence in $F_0$ shifts across turns, on the other hand, constitute two different dimensions of variation in $F_0$. The first dimension is concerned with how close speakers are to each other along the $F_0$ scale. The second dimension is linked to how accurately each speaker reproduces the other speaker's variations in $F_0$ over the reading-turn sequence. These two dimensions are, in principle, mutually independent: for example, it could be conceived that speakers espouse each other's changes in $F_0$ from one turn to the next whilst remaining at the same distance from each other on the $F_0$ scale. Our data, however, reveal that convergence between speakers occurred on both dimensions simultaneously, and that a greater amount of convergence in mean $F_0$ was associated with a greater amount of convergence in $F_0$ changes across turns.

We also found that convergence between speakers fell into place from the beginning of the joint reading task. This applied to both convergence in mean $F_0$ and convergence in $F_0$ shifts across turns. In the latter case in fact, the $\delta$ measure appeared to deviate from zero to a greater extent over the first part relative to the second part of the reading-turn sequence (see Fig 3). These results are at variance with a conventional view of convergence as a phenomenon that gradually builds up over the course of a speech production task (see [10, 54] for schematized representations of this conventional view). In contrast to this view, our results indicate that convergence in both mean $F_0$ and $F_0$ shifts across turns can be performed very quickly and as soon as speakers start interacting with each other. A potential limitation of our work relates to the fact that our pairs of speakers already knew each other, since the question may be raised whether familiarity between speakers may have contributed to facilitating convergence in $F_0$. However, in the only study known to us on the potential links between familiarity and convergence, Pardo and colleagues [55] found that perceived convergence between college roommates did not differ over the course of the academic year. Thus, the available experimental evidence does not point to an increase in phonetic convergence with increased familiarity.

Another significant outcome of this work is that speakers converged to a greater extent towards each other (as measured by $F_0$ shifts across turns) when they were already close to each other (as measured by overall proximity in mean $F_0$). This is at odds with an approach to convergence according to which speakers move towards a target that is halfway between them along one or several phonetic dimensions, with the implication that the speakers deviate more from their respective initial positions when these positions are further apart (see [36, 56], among others). Our results are more consistent with a different view, in which speakers engaged in a verbal interaction tend to become more phonetically alike when they already sound more like each other at the outset. It may indeed be assumed that phonetic convergence towards the interlocutor will be facilitated when that interlocutor's speech sounds are more within the range of the speaker's own, long-established, articulatory maneuvers [4].

Our data can be accounted for by means of a new, dynamical model of $F_0$ control based on three main assumptions. The first assumption is that speakers compute and store in memory a measure of mean $F_0$ in their interlocutor's speech over the interlocutor's speaking turn. This entails speakers' being able to abstract mean $F_0$ from the potentially large up-and-down $F_0$ excursions that the interlocutor may perform throughout the turn. The second assumption is that mean $F_0$ associated with the speaker's upcoming turn $t + 1$ is set as a function of both the speaker's mean $F_0$ in her/his last turn ($t - 1$), and the interlocutor's perceived mean $F_0$ in the ongoing turn $t$. The third assumption is that the interlocutor's contribution to setting the speaker's mean $F_0$ has a weight that is larger when the speaker and interlocutor are closer to each other on the $F_0$ dimension.

Our proposed account differs in several important respects from current models of $F_0$ convergence between speakers, such as the one exposed in [9]. First, these models appear to be agnostic as to the size of the time window over which the interlocutor's $F_0$ may be integrated, whereas we contend that this time window extends over one speaking turn. Second, we do not regard $F_0$ convergence as stemming from a perceptuo-motor recalibration mechanism, by virtue of which changes in a speaker's sensory targets occur as a result of that speaker being exposed to another speaker's voice. Rather, in our account, the two speakers are speaking to a common tune, i.e. the target mean $F_0$ for an upcoming turn is established by them in a joint manner. In other words, instead of conceiving $F_0$ convergence as a shift in each speaker's $F_0$ under the influence of an external speech input, we suggest that it is the product of a two-speaker shared sensory-motor plan. Finally, our model sets limits to $F_0$ convergence, which we expect to apply to a greater extent to speakers whose voices already resemble each other more.

Joint reading aloud is but one instance of a wide repertoire of behaviors today referred to as joint action. Joint action has been defined as a social interaction whereby two or more individuals coordinate their actions in space and time to bring about a change in the environment [25]. In this domain, a central issue is to what extent joint action is the result of joint planning, and entails using shared task representations and sensory-motor goals [26]. To our knowledge, our results provide the first piece of experimental evidence for convergence in $F_0$ as stemming from the use of shared representations and sensory-motor plans in a joint speech production task.

## Supporting information

**S1 Fig. Interface for the voice transformation software.** Top-left panel: global parameters with, from top to bottom: toggle Audio, set recording index, allow cross-talk, set tranformation's phase angle (0 or $\pi$), amplitude and period. Top-right panel: visual indicators monitored during the task. Audio signal and current transformation value for participants A and B are shown on the left and right respectively, with the current turn number in the center, and the recording indicator at bottom. Bottom panel: commands to control the task. Left: pushbuttons triggering audible instructions to participants for the 4 parts of the task (silent reading, practice text, first repetition of text, second repetition text. Right: pushbuttons to initialize the task, start and stop recording in green, red and gray respectively.
(PDF)

**S1 Text. Text read by the participants.** For participant A (version shown here), odd-numbered turns are in boldface and are to be read aloud while even-numbered turns are in gray color and are to be listened to in the partner's voice. For participant B (not shown), odd-numbered turns are in gray color, and even-numbered turns in boldface. Turn boundaries and turn numbers, added here for reference, were not shown in the version given to both participants.
(PDF)

## Acknowledgments

We are grateful to two anonymous reviewers for helpful comments and suggestions, and to Amandine Michelas, Robert Espesser and Silvain Gerber for technical assistance and fruitful discussions. We thank Fabienne Alibeu and Barbara Levy for help in the recruitment and testing of the participants. We also thank www.travaux.com for allowing us to use the text in the experiment.

## Author Contributions

**Conceptualization:** Vincent Aubanel, Noël Nguyen.

**Data curation:** Vincent Aubanel.

**Formal analysis:** Vincent Aubanel, Noël Nguyen.

**Funding acquisition:** Noël Nguyen.

**Investigation:** Vincent Aubanel, Noël Nguyen.

**Methodology:** Vincent Aubanel, Noël Nguyen.

**Project administration:** Noël Nguyen.

**Resources:** Noël Nguyen.

**Software:** Vincent Aubanel.

**Supervision:** Noël Nguyen.

**Validation:** Noël Nguyen.

**Writing – original draft:** Vincent Aubanel, Noël Nguyen.

**Writing – review & editing:** Vincent Aubanel, Noël Nguyen.

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
