## [Decision Letter · Decision Letter 0]

27 Dec 2019

PONE-D-19-26013

Speaking to a common tune: between-speaker convergence in voice fundamental frequency in a joint speech production task

PLOS ONE

Dear authors,

I have now received two clear and detailed assessments of the manuscript you submitted to PLOS One entitled "Speaking to a common tune: between-speaker convergence in voice fundamental frequency in a joint speech production task". While both reviewers consider this paper to represent an interesting contribution to the field, the first reviewer in particular raised a number of major issues regarding the description and analysis of the data. I agree with this reviewer' that the findings should be contextualized more precisely, and that the conclusions reached in some of the statistical analyses may not have been warranted. Taken together, addressing these and all other suggestions may require carrying out new analyses and reworking several sections of the text. However, I am convinced that addressing the reviewers' comments will improve the quality of your manuscript. I therefore hope you will consider resubmission. I look forward to receiving your revised manuscript.

We would appreciate receiving your revised manuscript by January 24. To enhance the reproducibility of your results, we recommend that if applicable you deposit your laboratory protocols in protocols.io, where a protocol can be assigned its own identifier (DOI) such that it can be cited independently in the future. For instructions see: http://journals.plos.org/plosone/s/submission-guidelines#loc-laboratory-protocols

We look forward to receiving your revised manuscript.

Kind regards,

Francisco José Torreira, Ph.D.

Academic Editor

PLOS ONE

Journal Requirements:

Additional Editor Comments:

l. 9: "speaker's mental lexicon": Perhaps you could use 'mental sound representations' or something similar, as it is not obvious whether/how phonetic convergence may affect the structure of the lexicon beyond phonetic and phonological processing.

l. 14: "indexical information associated with the speaker’s age, gender, and/or emotional state": Paralinguistic uses of F0 have been claimed to play communicative roles (cf. Ohala and Gussenhoven's paralinguistic uses of their biological codes).

l.57. Consider dropping "have" for consistency in the use of past tenses.

l. 85: "to what extent" this projects some kind of comparison later.

l.102: Consider dropping "the" in "the F0".

l.116: "that participants read aloud alternatively": it'd be useful for the reader to have an approximate idea of how long the task was.

l. 130: "achieve a turn". Not clear what is meant by "achieve" here.

l. 184: "following [13] study: Please check grammar.

l. 264: "7.5 turns": 75 turns

l. 303: "in the immediately preceding turn.". Since m3 performed better than m1 and m2, shouldn't turn be "preceding turns"?

l. 306: Consider dropping the word "even".

l. 316: "a tendency that was found to establish from": Please check grammar.

Reviewers' comments:

Reviewer's Responses to Questions

**Comments to the Author**

1. Is the manuscript technically sound, and do the data support the conclusions?

Reviewer #1: Yes

Reviewer #2: Yes

2. Has the statistical analysis been performed appropriately and rigorously? 

Reviewer #1: Yes

Reviewer #2: No

3. Have the authors made all data underlying the findings in their manuscript fully available?

Reviewer #1: Yes

Reviewer #2: No

4. Is the manuscript presented in an intelligible fashion and written in standard English?

Reviewer #1: Yes

Reviewer #2: Yes

5. Review Comments to the Author

Reviewer #1: In this study, the authors investigate pitch adaptation, the tendency for two speakers to converge towards similar pitch patterns over the course of a conversation (in this study during a scripted dialogue). They found that speakers imitated each other's changes in F0 across turns and that speakers were both able to perceive their partner’s changes in F0 and imitate these variations. The paper is clear and well written. The statistical analyses and experimental design are appropriate. The approach taken to study pitch convergence is novel and will be valuable to the expert reader. Below some suggestions for improvement of the manuscript.

Abstract :

1/ « F0, in spite of the major role played by F0”: instead of F0 (parameter), maybe rather use “pitch” (perception)

2/ “we asked to what extent two speakers converge towards each other with respect to variations in F0 in a scripted dialogue.”: investigate instead of ask

Title:

3/ “Speaking to a common tune: between-speaker convergence in voice fundamental frequency in a joint speech production task” the authors may just use the term pitch convergence to avoid the need of clarifying “voice” in “voice fundamental frequency”

Introduction

4/ “It is thought to contribute to setting up a conversational common ground between speakers and to facilitate mutual understanding “: the authors may add here the other effect/role of convergence in strengthening relationship. This is of importance as the participants in this study knew each other prior to the experiment. It may be assumed that some form of phonetic convergence is already set in place in other contexts of social interaction they are used to engage in.

5/ “As is well known,” maybe remove not necessary

6/ “The results, however, have shown important discrepancies both across and within studies,” ok but there are some consistencies of global pitch convergence (f0 median/ range) which has been well described in the literature. The authors could here be more specific about what would have led to inconsistencies across studies; what exactly was measured of f0? what temporal span was used to assess f0 changes across studies? how much time was given to the participants to potentially measure pitch convergence? How familiar were the speakers before they engaged in the experimental task?

7/ “Our main objective was to contribute to better characterizing the size of convergence effects in F0 in both the temporal and frequency domains. More specifically, we aimed to experimentally determine whether and if so to what extent convergence in F0 between human speakers extends across turns. We also sought to establish how accurately speakers may imitate changes in their partner's F0 that are both limited in magnitude and spread over large intervals.”: The objectives could come earlier in the introduction to highlight the novelty of the work. Otherwise it may not be clear to the reader in what way this study advances current knowledge and measurement of pitch convergence.

8/ “This framework could explain why speakers perform with outstanding accuracy with little training at uncommon speech tasks such as shadowing [8] or synchronous speech [35].” “with little training” could be misleading. Yes these are uncommon speech tasks and speakers are not trained for these specific tasks however the fact that they are consistently engaged in social interaction in their day-to-day activities make them expose to others’ speaking styles and therefore they may engage in pitch convergence on a daily basis.

Materials and methods

9/ The sections participants and procedure could be moved before the experimental design section to set the context for the experimental design.

10/ “communicated with each other using microphones and headphones.”: for the reader’s interest, the author could specify here the reason why the speakers were put in separate rooms (e.g. not to rely on non-verbal cues to communicate).

11/ “the participants did not notice that their partner's voice had been artificially manipulated.” How did the authors control for that? Post questionnaire? Also, what was the lag between speakers’ turns?

12/ For that reason, we recruited female participants: it is important to state this in the inclusion criteria and maybe add this as a limitation to the work as you may have measured here the effect of gender (male vs. female) or gender-mixed pair (female-female vs female-male) on adaptation capacity.

13/ “We chose our sample size following [13] study, where F0 convergence effects were observed with cohorts of 24 participants exposed to F0 values varying from 196 to 296 Hz (for female 185 voices), that is, a variation of 714 cents. Given our planned variation of 400 cents, that 186 is, about half of that in [13], we estimated that we would need at least twice as many 187 participants to observe an F0 convergence effect. .. we provisioned for an additional one third of participants, resulting in planned participants.” A power analysis maybe?

14/ “Participants came in pairs to the lab with the requisite that pair members knew each other well and had an age difference of no more than ten years. Both the familiarity and the age difference criteria were expected to facilitate the participants' joint accomplishment of the reading task.” A) This goes back to my previous comments that “little training” may not be appropriate in the context of this study, specifically when the authors ensured that the participants knew each other well to facilitate joint accomplishment; B) how did you control for similar familiarity across pairs? C) there may be a bit of circularity here. The results on the ability to both perceive slight changes in f0 and imitate F0 changes are to be taken in the context that the speakers were selected according to their familiarity. In this context, the results of them being able to quickly hear f0 changes and imitate are highly predictable and therefore it could be argued that the found convergence is just the result of pairing familiar individuals together.

15/ “median F0 value in the channel of the participant that had spoken during that turn” why not looking at the range too?

Results

16/ “We examined to what extent participants converged towards each other in mean F0, by calculating the correlation in mean F0 across pairs of participants. Over the entire duration of the task, this correlation was found to be significantly positive (r = 0:45; p < 0:02, see Fig 3a)”. Providing that the authors measure f0median over the turn and that F0 median per turn tend to decline over the course of a paragraph, how can the authors ascertain that the speakers were adapting to each other and that the found correlation is not a by-product of global declination naturally used by both readers in the context of text reading (in this global context - larger temporal span)?

17/ “showing that co-participants who were closer to each other in mean F0 tended to more closely imitate each other's perceived shifts in F0 from one turn to the next” yes and this should be contextualised as per my previous comment of using same-gender pair and concluding on ability to adapt.

18/ “as soon as each speaker starts being exposed to the other speaker's voice.” This could be misleading provided that the speakers are used to each other’s speaking style and know each other well.

19/ The authors could emphasize the novelty of their approach of pitch convergence (experimental design and measure), and may make it a study goal in order to highlight the contribution of their work to the literature.

Reviewer #2: My complete review is included as an attachment.

Re: question 3 above, the authors state that data will be available upon acceptance, but do not give the specified information as requested as to why this is the case.

6. PLOS authors have the option to publish the peer review history of their article (what does this mean?). If published, this will include your full peer review and any attached files.

Reviewer #1: No

Reviewer #2: No

---

## [Author Response · Author response to Decision Letter 0]

4 Feb 2020

The responses to the reviewers' and editor's comments are in the document entitled <Response to Reviewers.pdf>.

---

## [Decision Letter · Decision Letter 1]

10 Apr 2020

Speaking to a common tune: between-speaker convergence in voice fundamental frequency in a joint speech production task

PONE-D-19-26013R1

Dear Dr. Nguyen,

We are pleased to inform you that your manuscript has been judged scientifically suitable for publication and will be formally accepted for publication once it complies with all outstanding technical requirements. Following the recommendation of Reviewer #2, I encourage you to have your manuscript proofread by a native speaker if possible in order to improve the readability of the text.

With kind regards,

Francisco José Torreira, Ph.D.

Academic Editor

PLOS ONE

Additional Editor Comments (optional):

Reviewers' comments:

Reviewer's Responses to Questions

**Comments to the Author**

1. If the authors have adequately addressed your comments raised in a previous round of review and you feel that this manuscript is now acceptable for publication, you may indicate that here to bypass the “Comments to the Author” section, enter your conflict of interest statement in the “Confidential to Editor” section, and submit your "Accept" recommendation.

Reviewer #1: All comments have been addressed

Reviewer #2: (No Response)

2. Is the manuscript technically sound, and do the data support the conclusions?

Reviewer #1: Yes

Reviewer #2: Yes

3. Has the statistical analysis been performed appropriately and rigorously? 

Reviewer #1: Yes

Reviewer #2: Yes

4. Have the authors made all data underlying the findings in their manuscript fully available?

Reviewer #1: Yes

Reviewer #2: Yes

5. Is the manuscript presented in an intelligible fashion and written in standard English?

Reviewer #1: Yes

Reviewer #2: Yes

6. Review Comments to the Author

Reviewer #1: (No Response)

Reviewer #2: My concerns with the previous version have been substantially addressed. The manuscript should be proofread by a native speaker of English, since there are still a few usage errors to be corrected.

7. PLOS authors have the option to publish the peer review history of their article (what does this mean?). If published, this will include your full peer review and any attached files.

Reviewer #1: No

Reviewer #2: No

---

## [Editor Report · Acceptance letter]

16 Apr 2020

PONE-D-19-26013R1 

Speaking to a common tune: between-speaker convergence in voice fundamental frequency in a joint speech production task 

Dear Dr. Nguyen:

I am pleased to inform you that your manuscript has been deemed suitable for publication in PLOS ONE. Congratulations! Your manuscript is now with our production department. 

With kind regards,

on behalf of

Dr. Francisco José Torreira 

Academic Editor

PLOS ONE